# Peer review of "Integrative Multi-OMICs Identifies Therapeutic Response Biomarkers and Confirms Fidelity of Clinically Annotated, Serially Passaged Patient-Derived Xenografts Established from Primary and Metastatic Pediatric and AYA Solid Tumors"

_cancers, 2022, doi:10.3390/cancers15010259_

Round 1

Reviewer 1 Report

Highly relevant and important paper, extensive work, but to me short of conclusions on the feasibility in directly benefitting patients (in a practical manner). Please extend and see other comments below.

Figure 3B, figure legend or figure labels should be corrected, P0 seems to be missing but is discussed in the legend as well as the text. (I understand this is a resection of HT72 at a later stage, but P0 is used in other figures for HT77, so it is unclear why it’s missing in this figure. Please correct

Cdk4 and cdk6’s role in immune regulation is discussed as potential explanation for their decrease over time within the PDKs (line565). Authors should explain how this should be viewed in light of using NSG mice.

Figure 14 and S14, colour labelling, e.g. colour red for 3 different things is not ideal, especially not if printed black and white, use of different colour scheme that isn’t overlapping is recommended

Authors should comment on the feasibility implementing this in the future to clinical praxis (timelines from biopsy to analysis) and if their PDX time line is (on average) sufficient to guide the patient treatment in a timely manner. (i.e. expand on line 814-821 information gained from PDX too late for patient or how could it have benefitted the patient, in effect elaborating on the discussion (line 954 ). Could the information you gained from the pdx (or a more extensive one) predict that previous treatments would fail to impair tumour growth sufficiently? What about the other patients?

(as in conclusion it is stated that model can benefit patients, which I do not want to disagree with per se, but its questionable if the timeline at present is sufficient to allow this or what would need to done to achieve it)

Haven’t commented on success rate/engraftment rate of their PDX’s. Table/Figure or information on engraftment/success rate of stablishing a PDX for the different tumour types is needed, please add.

Author Response

Reviewer 1:

Comments and Suggestions for Authors

  1. Highly relevant and important paper, extensive work, but to me short of conclusions on the feasibility in directly benefitting patients (in a practical manner). Please extend and see other comments below.

We thank the reviewer to point the clinical significance which was not clear in previous version. We have modified our text to make it clearer. We believe that a major benefit of expanded pediatric and AYA PDX panels is for future patients and not necessarily for the patient from which the PDX was derived.  The field in pediatric and AYA solid tumors is at the point where scientists are generating efficacy and safety data to provide rationale for earlier intervention in future patients. However, there will be occasions where there is sufficient tumor tissue that allows development and characterization of a PDX in sufficient time to collect data that could provide information to the clinician in terms of clinical care, for patients with aggressive sarcomas can live for several years after diagnosis and treatment. Depending on the clinical situation, tumors are sequenced either upfront or following resection. In the future, we envision the molecular signature and the clinical history of a new patient will be compared to efficacy data as well as the efficacy/toxicity/adaptive response data we have already obtained from a previously molecularly characterized PDX panel that corresponds closely to the patient of interest. These data could help guide decision making for the patient. However, there will be cases where the PDX made from that same patient can be used to provide additional rationale to the physician regarding therapy. In discussions with our clinical team who are coauthors on this manuscript, they emphasize the efficacy data from a PDX would not be the defining component of a clinical decision but could certainly help increase confidence of a decision. As an example, in the metastatic Wilms tumor case described in this study (PDX HT120), this model was developed and characterized while the patient was still being treated.  Efficacy data from this model did in fact provided additional supportive data in a clinical decision for this patient, and a manuscript regarding this this study is under preparation. Since the submission of this manuscript this patient has unfortunately passed away and we have updated disease status in Table 1 and Table S1.

                Additionally, it should be emphasized that we have developed PDX models from - treatment-naïve and pre-treated PDXs. Treatment-naïve PDX models can help provide knowledge on the molecular landscape that exists prior to any first-line treatment. Pretreated PDXs can help give insight into tumor adaptation and possible therapeutic options for the next phase of therapy. Such information can help prioritize targeted therapies that can be implemented early on when the tumor burden is lower. Please see comment 4 below for more details. Additionally, we have expanded on this concept in the conclusions section of the manuscript. Please see page 41, lines 1097-1112. “A major benefit of such study is to expand pediatric and AYA PDX panels for future patients and not necessarily for the patients from which the PDX was derived. The field in pediatric and AYA solid tumors is at the point where scientists are generating efficacy and safety data to provide rationale for earlier intervention in future patients. Depending on the clinical situation, tumors are sequenced either upfront or following resection. In the future, we envision the molecular signature and the clinical history of a new patient will be compared and aligned to similar data sets of previous patient(s) from which PDX(s) was also generated. While efficacy data from a PDX would not be the defining component of a clinical decision, it could provide supportive data to help guide decision making for the newly diagnosed or relapsed patient.  As such, studies like this continue to be critical to identifying high yield targets and focusing clinical trial efforts on innovative therapeutics most likely to benefit these underserved pediatric populations. With that said, however, as more PDX models are developed and data is linked to the corresponding patient’s clinical history outcomes, these models may eventually become a more prominent and accepted component of clinical decision making.”

  1. Figure 3B, figure legend or figure labels should be corrected, P0 seems to be missing but is discussed in the legend as well as the text. (I understand this is a resection of HT72 at a later stage, but P0 is used in other figures for HT77, so it is unclear why it’s missing in this figure. Please correct

We thank the reviewer to point out the errors and have corrected Figure 3B per your suggestion from originally submitted manuscript.

  1. Cdk4 and CDK6’s role in immune regulation is discussed as potential explanation for their decrease over time within the PDKs. Authors should explain how this should be viewed in light of using NSG mice.

We apologize it was not clearly written in the previous version. We were actually referring to the different functions of CDK4 and CDK6 as demonstrated by Dai et al. It was not our intent to infer any type of immunoregulation in the immunodeficient NSG background. Therefore, to avoid confusion, we have simplified the text in page 17, lines 637-639 to the following “Differential expression of CDK4 and CDK6 was evident in the OS PDXs. Dai et al previously demonstrated that CDK4 and CDK6 can have distinct biological functions that regulate cancer growth and metastasis [ 80].

  1. Figure 14 and S14, colour labelling, e.g. colour red for 3 different things is not ideal, especially not if printed black and white, use of different colour scheme that isn’t overlapping is recommended

We thank the reviewer for this suggestion and have switched to a more clear color scheme for Figure 14 and Figure S14. The color shades are changed and now it is distinctive between different levels.

  1. Authors should comment on the feasibility implementing this in the future to clinical praxis (timelines from biopsy to analysis) and if their PDX time line is (on average) sufficient to guide the patient treatment in a timely manner. (i.e. expand on line 814-821 (from originally submitted manuscript) information gained from PDX too late for patient or how could it have benefitted the patient, in effect elaborating on the discussion (line 954 in the originally submitted manuscript ). Could the information you gained from the PDX (or a more extensive one) predict that previous treatments would fail to impair tumor growth sufficiently? What about the other patients? (as in conclusion it is stated that model can benefit patients, which I do not want to disagree with per se, but its questionable if the timeline at present is sufficient to allow this or what would need to done to achieve it).

We gratefully thank the reviewer to help us identify the clinical importance of this work. See comments above in our response to comment # 1 and page 41, lines 1097-1112 in the conclusions section of the manuscript.

To expand on what we stated in lines 814-821 from originally submitted manuscript and in line 954 in the originally submitted manuscript we have included the following statements in the conclusions section of the manuscript on page 41, lines 1097-1112: “A major benefit of such study is to expand pediatric and AYA PDX panels for future patients and not necessarily for the patients from which the PDX was derived. The field in pediatric and AYA solid tumors is at the point where scientists are generating efficacy and safety data to provide rationale for earlier intervention in future patients. Depending on the clinical situation, tumors are sequenced either upfront or following resection. In the future, we envision the molecular signature and the clinical history of a new patient will be compared and aligned to similar data sets of previous patient(s) from which PDX(s) was also generated. While efficacy data from a PDX would not be the defining component of a clinical decision, it could provide supportive data to help guide decision making for the newly diagnosed or relapsed patient.  As such, studies like this continue to be critical to identifying high yield targets and focusing clinical trial efforts on innovative therapeutics most likely to benefit these underserved pediatric populations. With that said, however, as more PDX models are developed and data is linked to the corresponding patient’s clinical history outcomes, these models may eventually become a more prominent and accepted component of clinical decision making.”

  1. Haven’t commented on success rate/engraftment rate of their PDX’s. Table/Figure or information on engraftment/success rate of stablishing a PDX for the different tumor types is needed, please add.

We have added the information from the table below to the Materials and Methods section (page 5, lines 181 -193) The first priority for tumor samples at our institution is for diagnosis and specimens required for ongoing clinical trials. Generation of PDX is the next priority. Tissue quantity can vary greatly from patient-to-patient. The PDX characterized in this study are a subset of our larger pediatric and AYA PDX panel of rare tumors. Over the past several years, our success rate was defined as expansion of PDX at least through passage 2. In OS, 51.4% cases resulted in successful PDX generation. For RMS cases, 53.8% led to successful PDX generation, and 80% Wilms tumor cases gave rise to successful Wilms tumor PDXs. For the biopsy samples used for PDX generation, the surgeon’s expertise and tumor scans are used to identify viable tumor tissue. For resection samples used for PDX generation, the pathologists release viable tumor based on their expert analysis of the tissue sent by the surgical team. To transfer the tissue sample to the lab for further processing, MACS tissue storage medium was used (Miltenyi Biotec,cat # 130-1100-008).”

Reviewer 2 Report

Authors established PDX models for primary & metastatic pediatric and AYA solid tumors and performed multi-omics analysis to investigate therapeutic response.

Pediatric and specifically AYA is a distinct age group of interest that is not very well studied.  The research is very interesting and with PDX data it adds more value to the pre-clinical understanding of AYA.

Minor comments:

1.     AYA criteria are defined differently in different countries. So, it is recommended that the author define the age group inclusion in the introduction section.

2.     In lines 1-3, the authors mentioned, PDX mimics the molecular landscape of a patient's tumor. While this is true in most cases, there are many published and unpublished research that shows PDX does not always recapitulate the patient tumor characteristics and depends on cancer type. It is suggested that the author provide complete information.

3.     Patient tumors as well as PDX are highly heterogenous so a successful PDX study depends on the location of tissue resection etc. It would be clear to readers if the authors can provide some more information on tissue resection and sample preparation.

4.     It is mentioned BBsplit was used as a mouse read sorting tool in WGS sequence analysis but it is not clear if the same tool was used for removing RNA-seq reads. Also, did the authors use a human DNA capture kit for WGS?

5.     TMB inflation is reported in patients of different race/ethnicity. PDX models used in this study are mainly based on patients of the Caucasian race, so it would not always predict response to therapy. Such limitations should be mentioned.

6.     Line 430 says, unsupervised clustering was performed. Since there are many different unsupervised clustering methods, general terms must be explained with specific algorithms used for unsupervised clustering.

Major comments:

Experiments are performed very well. The manuscript is well-written and methodically sound but results section needs major revision. Figures and data are presented poorly, in a redundant manner, and not reader-friendly.

1.     The whole panel of Figure 1, analysis of congruence of genes and correlation can be summarized in 1 single plot with different plots. Please refer to other major genomic papers on how to concisely present such data.

2.     Fig 2, same as comment 1.

3.   Fig 3, there is no legend title. Readers must not struggle to find what values are represented.

4.     Again, in figure 3, all heatmaps can be summarized in 1 heatmap by faceting.

5.     Fig 6, summarize into 1 figure.

6.     Figure 7, it is very difficult to interpret the Venn diagram, it will hurt readers' eyes. It is highly recommended to use the UpSet plot in replacement of the Venn diagram. There is an R package you could use. Please check UpsetR in CRAN.

7.     You could add a new plot summarizing common and different pathways for all the passages and PDX.

8. Figure 14, the scale does not have a legend title.

Author Response

Reviewer 2:

Comments and Suggestions for Authors

Authors established PDX models for primary & metastatic pediatric and AYA solid tumors and performed multi-omics analysis to investigate therapeutic response.

Pediatric and specifically AYA is a distinct age group of interest that is not very well studied.  The research is very interesting and with PDX data it adds more value to the pre-clinical understanding of AYA.

Minor comments:

  1. AYA criteria are defined differently in different countries. So, it is recommended that the author define the age group inclusion in the introduction section.

We thank the reviewer for bringing up this point. Based on National Cancer Institute at the National Institute of Health in the United States adolescents and young adults are defined as individuals that are 15-39 years of age, individuals between 0-14 years of age are classified as children, and adult refers to anyone who is 40 years old or older. We have also included this age criteria between children and AYA in the introduction (page 3, lines 75-78) as follows ”While solid tumors are quite rare in this patient population, solid tumors represent approximately  60% of all malignant neoplasms observed in children (0-14 years), as well as adolescents  and young adults (AYA; 15-39 years) in the United States [3, 4].”

  1. In lines 1-3, the authors mentioned, PDX mimics the molecular landscape of a patient's tumor. While this is true in most cases, there are many published and unpublished research that shows PDX does not always recapitulate the patient tumor characteristics and depends on cancer type. It is suggested that the author provide complete information.

We agree that some PDXs do not always faithfully recapitulate the original patient tumor characteristics. We mentioned this point in the discussion (page 40, lines 1045-1053) where we stated As previously mentioned in the introduction, a number of factors can contribute to the differences observed between the pairs of P0 tumor and passaged PDXs, such as clonal evolution, genetic drift, intra- and inter-tumoral heterogeneity, presence of human stroma and immune system in original tumors versus increased murine stromal content in passaged PDXs, insufficient amount of the original tumor sequenced, as well as lineage splitting due to tumors’ spatial heterogeneity during PDX propagation [16-19]. Additionally, it should also be noted that it is impossible to know if clonal drift or evolution that appears in some PDXs over time might also naturally occur in that same patient at some point in time over the course of their disease. “

We now expand upon this issue in more detail. We agree with the reviewer that we need to mention this earlier in the manuscript. However, we don’t see this in lines 1-3 as the reviewer indicates but anticipate that the reviewer was suggesting lines 101-103 in the originally submitted manuscript. Thus, we have provided the following revisions in the introduction (page 3, lines 104-120):

“There are some cases where the PDX model differentiates from the original human tumor. This can be attributed to sampling bias of genetically heterogenous tumors that may inadvertently occur [14]. The dissected piece of the tumor for PDX development may not fully represent the molecular landscape of the entire tumor [15] which, ultimately, may cause further clonal evolution and genetic drift compared to the original tumor [15-19]. It has also been reported that PDXs developed from advanced tumors have higher take-rate than those developed from less aggressive tumors (at early stages of disease progression) which may not fully mimic the full range of molecular landscape present in the original tumor [14, 20-23]. Additionally, since PDXs lack the immune system as well as the vasculature and stromal components from the host, tumors that predominately rely on stromal components for growth may not provide good fidelity between the PDX and the original tumor [15]. 

Despite some PDX models not truly recapitulating the molecular characteristics of the original human tumor, overall concordance between key oncogenes in PDXs and their respective original human tumor are evident in the majority of PDXs shown here as well as other comprehensive, large-scale genomic characterization studies conducted in PDXs encompassing a variety of solid tumors such as carcinomas, sarcomas, and melanomas [24].”

  1. Patient tumors as well as PDX are highly heterogenous, so a successful PDX study depends on the location of tissue resection etc. It would be clear to readers if the authors can provide some more information on tissue resection and sample preparation.

We thank the reviewer for this great recommendation. We agree with the comment regarding patient tumor heterogeneity and attempting to obtain tumor samples that encompass a good representation of molecular heterogeneity. We have added the following information to the Material and Methods section, (page 5, lines 188-193). ”For the biopsy samples used for PDX generation, the surgeon’s expertise and tumor scans are used to identify viable tumor tissue. For resection samples used for PDX generation, the pathologists release viable tumor based on their expert analysis of the tissue sent by the surgical team. To transfer the tissue sample to the lab for further processing, we use MACS tissue storage medium was used (Miltenyi Biotec,cat # 130-1100-008).”

  1. It is mentioned BBsplit was used as a mouse read sorting tool in WGS sequence analysis but it is not clear if the same tool was used for removing RNA-seq reads. Also, did the authors use a human DNA capture kit for WGS?

We thank the reviewer for this very helpful feedback. DNA capture kits were not used.

We have provided more details: For HT72, HT77, HT87, HT96, HT74, and HT98 WGS was done at New York Genome Sequencing Center using TruSeq PCR-free preparation, (page 5, lines 203-204). For HT120 and HT139 samples the WGS was performed at Indiana University School of Medicine Center for Medical Genomics where they used Illumina Nextera DNA Flex Library Prep Kit (page 5, lines 205-206). RNA-seq analysis was conducted at Quick Biology Inc where BBsplit function in the BBmap tool was utilized to remove the mouse reads (page 6, lines 272-273).

  1. TMB inflation is reported in patients of different race/ethnicity. PDX models used in this study are mainly based on patients of the Caucasian race, so it would not always predict response to therapy. Such limitations should be mentioned.

We thank the reviewer for the insight on this matter and have modified our text accordingly. We agree that racial/ethnic diversities may have an impact on therapeutic response as indicated by studies from Asmann et al., and Brawley et al. Notably, both of these studies highlight the importance of TMBs in predicting response to immunotherapies.

For instance, study by Asmann et al. (https://www.ncbi.nlm.nih.gov/pmc/articles/PMC7979755/), focuses on using TMBs as a biomarker for therapeutic response to immune-checkpoint inhibitors (PD-1 inhibitor). The study shows that TMBs are more inflated in African-American patients compared to Caucasian patients when patient-paired germline sequencing is not conducted but publicly-available databases are used instead.

Additionally, Brawley and group (https://www.ncbi.nlm.nih.gov/pmc/articles/PMC8577674/)  assessed in NSCLC how TMBs impact response to immunotherapies. In this study, they observed that in comparison to Caucasians, patients with African-American ancestry had higher TMBs, whereas, patients with Asian ancestry had lower TMBs.

As such, we are also interested in understanding how racial/ethnic diversity impacts TMBs and, ultimately, therapeutic response. As part of our Precision Genomics program, we are establishing a larger panel of pediatric solid tumor PDXs developed at our institution that encompasses more biological diversity such as various races, ethnicities, genders, age, and site of primary vs metastatic tumor.

For our current study, we focused on the first set of PDXs developed at the institution that were just by chance all derived from Caucasians. Since these models are propagated in immunodeficient mice we don’t anticipate a direct correlation between TMBs and therapeutic response to immunotherapies. However, we do see great merit in learning more about how racial/ethnic and other biological diversity may impact therapeutic response and, therefore, have elaborated on this in the discussion (pages 38-39, lines 982 -990) where we stated “It is important to note that, one limitation in our study is that by chance all of our PDXs were developed from Caucasian patients (Table 1) during the initial stage of program development  at our institution. It has been reported that racial and ethnic diversities can have an impact on tumor mutational burden as well as on response to immune-based therapies [162, 163]. Furthermore, tumor mutational burdens for races such as African-American patients have been reported to be inflated when publicly-available germline variant databases are used for filtering non-somatic polymorphisms [163] demonstrating the need to use patient-specific germlines.

  1. Line 430 (in the originally submitted manuscript) says, unsupervised clustering was performed. Since there are many different unsupervised clustering methods, general terms must be explained with specific algorithms used for unsupervised clustering.

Major comments:

Experiments are performed very well. The manuscript is well-written and methodically sound but results section needs major revision. Figures and data are presented poorly, in a redundant manner, and not reader-friendly.

  1. The whole panel of Figure 1, analysis of congruence of genes and correlation can be summarized in 1 single plot with different plots. Please refer to other major genomic papers on how to concisely present such data.

We thank the reviewer for this great suggestion and have combined the individual Jaccard plots we had for each P0-PDX cohort in the originally submitted manuscript in such a way that now all osteosarcoma PDXs are grouped together as a correlation heatmap (new Figure 1A in revised manuscript). Similar plots are made for rhabdomyosarcoma (new Figure 1B in revised manuscript) and for Wilms tumors (new Figure 1C in revised manuscript). Please see page 13, lines 532-537.

  1. Fig 2, same as comment 1.

We have corrected this as we have mentioned in our response above (comment 1) for Figure 1.  Please see page 14, lines 538-543 for the new Figure 2 in revised manuscript.

  1. Fig 3, there is no legend title. Readers must not struggle to find what values are represented.

We thank the reviewer for the helpful suggestion. We have Figure 3 legend title present on page 16, lines 598-599 where we state Distribution of copy number events involving COSMIC-associated genes in P0 tumors and corresponding PDX passages.” We have also included a title above the keys for each of the heatmaps which depicts copy numbers and their meanings (neutral, deletion, gains, or amplifications).

  1. Again, in figure 3, all heatmaps can be summarized in 1 heatmap by faceting.

We thank the reviewer for this suggestion. However, we prefer to keep this figure as is and not facet everything into one big plot because our goal was to look at similarities and differences between passages within the PDXs not amongst PDXs.

  1. Fig 6, summarize into 1 figure.

We thank the reviewer for this suggestion and have corrected Figure 6 from originally submitted manuscript. Please see page 21, lines 685- 690 for the new Figure 6 in revised manuscript.

  1. Figure 7, it is very difficult to interpret the Venn diagram, it will hurt readers' eyes. It is highly recommended to use the UpSet plot in replacement of the Venn diagram. There is an R package you could use. Please check UpsetR in CRAN.

We have converted the Venn diagram figures in the originally submitted manuscript (Figure 7, Supplemental Figures S11 and S12) into Upset plots per your suggestion. Please see page 23, lines 702-708 for the new Figure 7 and refer to Figures S11, 12 in supplemental figures section of revised manuscript.

  1. You could add a new plot summarizing common and different pathways for all the passages and PDX.

Please see attached excel table.

We thank the reviewer for this suggestion and followed it to summarize them in the form of a supplemental table (Table S7 in the revised manuscript). We have also added the following text “Based on the RRPA data, there is evidence that a number of actionable cancer-associated pathways are indeed activated in all of the PDX samples. However, there are relative differences in the activation level of specific pathways across the PDX panel (Table S7). Whether these subtle differences in pathway activation equate to differential therapeutic responses in each PDX warrants further investigation (page 35, lines 888-892).

  1. Figure 14, the scale does not have a legend title.

We have corrected this and added a scale title in the revised manuscript’s new Figure 14 (page 34, lines 870 -876).

Reviewer 3 Report

Solid tumors of pediatric cancers represent a large proportion of malignant neoplasms in patients. It is important to develop safe and efficient therapeutic strategies to facilitate the treatment of pediatric cancers with solid tumors. In this manuscript, the authors used the samples from osteosarcoma (OS), rhabdomyosarcoma (RMS), and the Wilms tumor to establish a multi-OMICs pipeline to identify the biomarkers that can be beneficial to the therapies that are guided by precision genomics. PDXs were developed from the original tumor with different passages. DNA and RNA sequencing were done by using p0 and the corresponding passages. Then the authors did intensive bioinformatics analysis including somatic whole genome analysis, copy number variation, single nucleotide variation, gene expression, and enrichment analysis. Some genes were also compared at the protein level. Overall, the manuscript was interesting and the datasets were useful, however, the manuscript can be improved and some issues need to be addressed.

1.    To ensure reproducibility, in the method sections, the authors should specify the parameters that were used for the bioinformatics software, for example, BWA, BBsplit, BBtools, CNV kit… Alternatively, the authors could release the code that was used to run these programs.

2.    Section 3.3 was not clear to be, what kind of genome-wide analysis was performed. And what was the result of it? What does it mean by “similarities and differences in DNA sequences of the P0 versus the respective passages”

3.    Figure 1 to figure 3, I wonder if the authors could re-organize these figures by three panels (instead of 8 panels). The authors can put ABCD for OS samples together in one panel (A), the E for RMS samples in one panel (B), and FGH for Wilms tumor samples in the last panel. In this way, the readers do not need to go back and forth to find the samples that belong to the same tumor group.

4.    Line 489, it was not clear how the authors selected the 25 genes, was it based on their expression in different tumors?

5.    Lines 489-528, there was too much information in this section, introducing the function of each gene was disruptive, the authors should consider simplifying the content in this section.

6.    In figure 3 and other figures after this, why there was no p0 data for HT77?

7.    In figure 3, why the gene list was different for HT96 from the lists for HT72/HT77/HT87? (ATM was only present in HT96 but not in others)

8.    Section 3.5, because for many genes, the protein levels were not correlated with the copy numbers. I wonder if the RNA levels were correlated with the copy numbers of these genes.

9.    Figure 4, why p0 data was not included in this gel?

10. Figure 5, were the numbers qualified by per-genome size? How was the result normalized between different datasets?

11. How was the data for figure 8 related to figure 5E? For example, in figure 5E, there were no single base substitutions in HT72 and HT87, but in figure 8, there were many transversions and transitions.

12. Section 3.8, what kind of somatic variants were used for this analysis? Or all the variants were added together?

13. Generated raw data should be publicly available. The authors should deposit the raw data to a public database such as NCBI.

Author Response

Reviewer 3:

Comments and Suggestions for Authors

Solid tumors of pediatric cancers represent a large proportion of malignant neoplasms in patients. It is important to develop safe and efficient therapeutic strategies to facilitate the treatment of pediatric cancers with solid tumors. In this manuscript, the authors used the samples from osteosarcoma (OS), rhabdomyosarcoma (RMS), and the Wilms tumor to establish a multi-OMICs pipeline to identify the biomarkers that can be beneficial to the therapies that are guided by precision genomics. PDXs were developed from the original tumor with different passages. DNA and RNA sequencing were done by using p0 and the corresponding passages. Then the authors did intensive bioinformatics analysis including somatic whole genome analysis, copy number variation, single nucleotide variation, gene expression, and enrichment analysis. Some genes were also compared at the protein level. Overall, the manuscript was interesting and the datasets were useful, however, the manuscript can be improved and some issues need to be addressed.

  1. To ensure reproducibility, in the method sections, the authors should specify the parameters that were used for the bioinformatics software, for example, BWA, BBsplit, BBtools, CNV kit… Alternatively, the authors could release the code that was used to run these programs.

We thank the reviewer for this suggestion, which was not clearly written in the previous version. We have provided more information on the parameters used for bioinformatics analysis. Please see page 5, lines 202-218 where we state “WGS was performed at New York Genome Center (NYGC, New York, NY, USA) on the following PDXs: HT72, HT77, HT87, HT96, HT74, and HT98. For these samples, TruSeq PCR-free preparation was done. WGS on HT120 and HT139 was performed at IUSM Center for Medical Genomics core using the Illumina Nextera DNA Flex Library Prep Kit. For all P0 specimens and PDX samples, the sequencing depth/coverage was 90x. All SNV and CNV data is  uploaded to dbGaP database (https://www.ncbi.nlm.nih.gov/gap/). OS, RMS, and Wilms tumor cohorts were sequenced using NovaSeq 6000 Illumina platform to generate raw reads in FASTQ format. The raw sequencing data for the control sample, NA12878 was obtained from the International Genome Sample Resource and used as the normal reference human genome. Adapter sequences, primers and other unwanted sequences were first identified and trimmed using Cutadapt [32] and later filtered against mouse genome using BBsplit [33] from BBtools suite, to remove mouse-originated reads. The raw sequencing reads were aligned to the human reference genome hg38 with Burrows-Wheeler Aligner (BWA) [34]. The objective of the bioinformatics analyses was to compare the copy number variations (CNVs) and single nucleotide variations (SNVs) of each P0 versus respective PDX passages as well as compare within each set of PDX passages.”  Additionally, the R script codes will be submitted to github.

  1. Section 3.3 was not clear to be, what kind of genome-wide analysis was performed. And what was the result of it? What does it mean by “similarities and differences in DNA sequences of the P0 versus the respective passages”

We have clarified section 3.3 in terms of what kind of analysis was conducted by stating on page 11, lines 456 -458 “Genome-wide variant analysis was done to identify and compare the molecular signatures at the DNA level within each P0-PDX cohort.” 

  1. Figure 1 to figure 3, I wonder if the authors could re-organize these figures by three panels (instead of 8 panels). The authors can put ABCD for OS samples together in one panel (A), the E for RMS samples in one panel (B), and FGH for Wilms tumor samples in the last panel. In this way, the readers do not need to go back and forth to find the samples that belong to the same tumor group.

We thank the reviewer for this suggestion and changed the plots accordingly. For Figures 1 and Figures 2, we have combined the individual Jaccard plots we had for each P0-PDX cohort in the originally submitted manuscript in such a way that now all osteosarcoma PDXs are grouped together as a correlation heatmap (new Figure 1A or Figure 2A in revised manuscript). Similar plots are made for rhabdomyosarcoma (new Figure 1B or Figure 2B in revised manuscript) and for Wilms tumors (new Figure 1C or Figure 2C in revised manuscript). Please see pages 13 and 14. For Figure 3, we prefer to keep this figure as is and not facet everything into one big plot because our goal was to look at similarities and differences between passages within the PDXs not amongst PDXs. Please see page 16.

  1. Line 489 (in the originally submitted manuscript), it was not clear how the authors selected the 25 genes, was it based on their expression in different tumors?

We apologize it was not clear in the text. The reviewer is correct in that 25 COSMIC genes were selected/prioritized based on their expression and role in the tumor it was present in. We provide the rationale for this prioritization on page 15, lines 552-553. We state: “CNV profiles of 25 genes from the COSMIC database were prioritized based on their expression and the role they have in that particular cancer type,”. Appropriate literature has been cited for each prioritized gene set.

  1. Lines 489-528 (in the originally submitted manuscript), there was too much information in this section, introducing the function of each gene was disruptive, the authors should consider simplifying the content in this section.

We have now revised the section and deleted the gene functions in this section of the manuscript (page 15, lines 552-596).

  1. In figure 3 and other figures after this, why there was no p0 data for HT77?

We thank the reviewer for point it out, which was not clearly stated. HT72 and HT77 PDXs were derived from metastatic sites of the same pre-treated male OS patients at different time points during salvage therapy (please see page 16, lines 602-604). Therefore, the P0 from HT72 patient is the same P0 for HT77. We have now added in P0 to Fig 3B and other figures where sufficient tissue was available for analysis.

  1. In figure 3, why the gene list was different for HT96 from the lists for HT72/HT77/HT87? (ATM was only present in HT96 but not in others)

We thank the reviewer to point it out. As we have replied in the comment 4, the 25 genes were COSMIC genes selected/prioritized based on their expression and role in the tumor it was present in. We have states this as “CNV profiles of 25 genes from the COSMIC database were prioritized based on their expression and the role they have in that particular cancer type,” (page 15, lines 552-553).  For our analysis, if a gene had similar copy number (either neutral, gains, amplifications, or deletions) in the normal control sample (NA12878), P0, and its subsequent PDXs passages (P1-P2 or P3) then that gene was removed as there was no change. However, if a gene had copy number variations in P0 and/or any of its respective serially-passaged PDXs compared to the normal control (NA12878) than that gene was displayed in the heatmap. As such, if we use ATM as an example we see that in Fig. 3D ATM has a copy number gain present in P0, P1, P2, and P3 of HT96. This is because the gain was not observed in NA12878 normal control sample we filtered against. However, we do not see ATM copy number gain, amplification, or deletion in Figs 3A-C (HT72, HT77, and HT87) because the copy number change whether it was neutral, gain, amplification, or deletion was not different between NA12878 vs P0, P1, P2, and/or P3. To avoid confusion, we have included the following statement “It should be noted that not every gene will appear in each PDX cohort. As an example, ATM has a copy number gain present in P0, P1, P2, and P3 of HT96 that was not observed in NA12878 normal control sample that was used as the filter. However,  ATM copy number gain, amplification, or deletion in Figs 3A-C (HT72, HT77, and HT87) is not listed since the copy number change was not different between NA12878 vs P0, P1, P2, and/or P3,” (page 15, lines 520-573).

  1. Section 3.5, because for many genes, the protein levels were not correlated with the copy numbers. I wonder if the RNA levels were correlated with the copy numbers of these genes.

We thank the reviewer for this great suggestion. Taking your feedback into consideration we did look at whether the RNA/transcript levels correlated with copy number and/or protein expression of these genes.

We have included this information as a supplemental table (Table S6) and highlighted this concept on page 30, lines 811-819 where we stated the following “Whether the transcript levels correlated with copy number of genes and/or protein expression from really depended on the gene interest and which PDX was being investigated. For example, in HT72, HT77, and HT96 transcript levels and copy numbers for CCND3 were increased. Additionally, an increase in Cyclin D3 protein expression for HT72-P2, HT77-P2, and HT96-P2 was also evident (Fig. 4). For CDK6 increased transcript levels correlated within increased protein expression in HT72-P2 and HT87-P2 (Fig. 4). However, for other genes like TP53 transcript levels did not necessarily correlate with copy numbers and/or protein expression. Such discrepancies may be attributed to post-transcriptional and/or post-translational modifications.”

  1. Figure 4, why p0 data was not included in this gel?

We apologize it was not clear in the previous version. P0 samples are the original human tumor samples we received from the operating room. The sample size for P0 is very small and limited as sections of the sample are first prioritized to go to pathology and possibly for clinical sequencing and trial correlative studies. Our lab prioritized P0 samples from OS patients for serial- passaging of PDXs as well as whole-genome sequencing (WGS). If sufficient amount of P0 sample is still left following serial-passaging and WGS, then transcriptome analysis (RNA-seq) is prioritized. We have included an explanation for why P0 samples were absent in Fig. 4 (section 3.5 and in methods page 7, lines 285-286).

 Due to limited availability of P0 samples, they were prioritized for WGS and RNAseq.”

  1. Figure 5, were the numbers qualified by per-genome size? How was the result normalized between different datasets?

We have included the information on how the data was normalized. This is stated in Fig. 5 legend (pages 19-20, lines 671-673 ) “The data is normalized to total mappable variant counts specific to that particular sample. All data are normalized proportions to 10-4 as indicated by y-axis. The variant frequency for each PDX was normalized to sequencing depth.”

  1. How was the data for figure 8 related to figure 5E? For example, in figure 5E, there were no single base substitutions in HT72 and HT87, but in figure 8, there were many transversions and transitions.

Data in Figure 5 is based on variants on COSMIC genes only, whereas, Figure 8 was made based on variants in all genes. We have put this in the figure legends of Figure 5 (page 19, line 670) and Figure 8 (page 25, line 730).

  1. Section 3.8, what kind of somatic variants were used for this analysis? Or all the variants were added together?

This is referring to Data in Fig. 9. The variants are from COSMIC genes.

  1. Generated raw data should be publicly available. The authors should deposit the raw data to a public database such as NCBI.

We thank the reviewer for highlighting this point. We have already started the submission process of depositing our raw datasets/files to dbGaP (https://www.ncbi.nlm.nih.gov/gap/). Due to the large nature of the CNV and SNV datasets the uploading and submission is still in progress but will be completed prior to manuscript being published (page 42, lines 1148-1150).

Round 2

Reviewer 2 Report

In the revised manuscript, authors has addressed all the comments.